# `baller2vec`: A Multi-Entity Transformer For Multi-Agent Spatiotemporal Modeling

## Abstract

Multi-agent spatiotemporal modeling is a challenging task from both an algorithmic design and computational complexity perspective. Recent work has explored the efficacy of traditional deep sequential models in this domain, but these architectures are slow and cumbersome to train, particularly as model size increases. Further, prior attempts to model interactions between agents across time have limitations, such as imposing an order on the agents, or making assumptions about their relationships. In this paper, we introduce `baller2vec`[1], a multi-entity generalization of the standard Transformer that can, with minimal assumptions, *simultaneously and efficiently* integrate information across entities and time. We test the effectiveness of `baller2vec` for multi-agent spatiotemporal modeling by training it to perform two different basketball-related tasks: (1) simultaneously modeling the trajectories of all players on the court and (2) modeling the trajectory of the ball. Not only does `baller2vec` learn to perform these tasks well (outperforming a graph recurrent neural network with a similar number of parameters by a wide margin), it also appears to "understand" the game of basketball, encoding idiosyncratic qualities of players in its embeddings, and performing basketball-relevant functions with its attention heads.

## 1 Introduction

Whether it is a defender anticipating where the point guard will make a pass in a game of basketball, a marketing professional guessing the next trending topic on a social media platform, or a theme park manager forecasting the flow of visitor traffic, humans frequently attempt to predict phenomena arising from processes involving multiple entities interacting through time. When designing learning algorithms to perform such tasks, researchers face two main challenges:

1. Given that entities lack a natural ordering, how do you effectively model interactions between entities across time?
2. How do you *efficiently* learn from the large, high-dimensional inputs inherent to such sequential data?

Prior work in athlete trajectory modeling, a widely studied application of multi-agent spatiotemporal modeling (MASM; where entities are agents moving through space), has attempted to model player interactions through "role-alignment" preprocessing steps (i.e., imposing an order on the players) [1, 2] or graph neural networks [3], but these approaches may destroy identity information in the former case (see Section 4.2) or limit personalization in the latter case (see Section 5.1). Recently, researchers have experimented with variational recurrent neural networks (VRNNs) [4] to model

---

[1]All data and code for the paper are available at: <anonymized>.

the temporal aspects of player trajectory data [3, 2], but the inherently sequential design of this architecture limits the size of models that can feasibly be trained in experiments.

Transformers [5] were designed to circumvent the computational constraints imposed by other sequential models, and they have achieved state-of-the-art results in a wide variety of sequence learning tasks, both in natural language processing (NLP), e.g., GPT-3 [6], and computer vision, e.g., Vision Transformers [7]. While Transformers have successfully been applied to *static* multi-entity data, e.g., graphs [8], the only published work we are aware of that attempts to model multi-entity *sequential* data with Transformers uses four different Transformers to *separately* process information temporally and spatially before merging the sub-Transformer outputs [9].

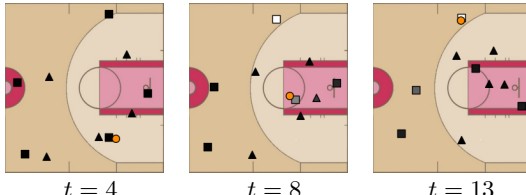

$t = 4$      $t = 8$      $t = 13$

Figure 1: After solely being trained to model the trajectory of the ball (⬤) given the locations of the players and the ball on the court through time, a self-attention (SA) head in `baller2vec` learned to anticipate passes. When the ball handler (■⬤) is driving towards the basket at $t = 4$, SA assigns near-zero weights (black) to all players, suggesting no passes will be made. Indeed, the ball handler did not pass and dribbled into the lane ($t = 8$). SA then assigns a high weight (white) to a teammate (□), which correctly identifies the recipient of the pass at $t = 13$.

In this paper, we introduce a *multi-entity* Transformer that, with minimal assumptions, is capable of *simultaneously* integrating information across agents and time, which gives it powerful representational capabilities. We adapt the original Transformer architecture to suit multi-entity sequential data by converting the standard self-attention mask matrix used in NLP tasks into a novel self-attention mask *tensor*. To test the effectiveness of our multi-entity Transformer for MASM, we train it to perform two different basketball-related tasks (hence the name `baller2vec`): (1) simultaneously modeling the trajectories of all players on the court (**Task P**) and (2) modeling the trajectory of the ball (**Task B**). Further, we convert these tasks into classification problems by binning the Euclidean trajectory space, which allows `baller2vec` to learn complex, multimodal trajectory distributions via strictly maximizing the likelihood of the data (in contrast to variational approaches, which maximize the evidence lower bound and thus require priors over the latent variables). We find that:

1. `baller2vec` is an effective learning algorithm for MASM, obtaining a perplexity of 1.64 on **Task P** (compared to 15.72 when simply using the label distribution from the training set) and 13.44 on **Task B** (vs. 316.05) (Section 4.1). Further, compared to a graph recurrent neural network (GRNN) with similar capacity, `baller2vec` is ~3.8 times faster and achieves a 10.5% lower average negative log-likelihood (NLL) on **Task P** (Section 4.1).
2. `baller2vec` demonstrably integrates information across *both* agents and time to achieve these results, as evidenced by ablation experiments (Section 4.2).
3. The identity embeddings learned by `baller2vec` capture idiosyncratic qualities of players, indicative of the model's deep personalization capabilities (Section 4.3).
4. `baller2vec`'s trajectory bin distributions depend on both the historical and current context (Section 4.4), and several attention heads appear to perform different basketball-relevant functions (Figure 1; Section 4.5), which suggests the model learned to "understand" the sport.

## 2 Methods

### 2.1 Multi-entity sequences

Let $A = \{1, 2, \ldots, B\}$ be a set indexing $B$ entities and $P = \{p_1, p_2, \ldots, p_K\} \subset A$ be the $K$ entities involved in a particular sequence. Further, let $Z_t = \{z_{t,1}, z_{t,2}, \ldots, z_{t,K}\}$ be an *unordered* set of $K$ feature vectors such that $z_{t,k}$ is the feature vector at time step $t$ for entity $p_k$. $\mathcal{Z} = (Z_1, Z_2, \ldots, Z_T)$ is thus an *ordered* sequence of sets of feature vectors over $T$ time steps. When $K = 1$, $\mathcal{Z}$ is a sequence of individual feature vectors, which is the underlying data structure for many NLP problems.

We now consider two different tasks: (1) sequential entity labeling, where each entity has its own label at each time step (which is conceptually similar to word-level language modeling), and (2) sequential

labeling, where each time step has a single label (see Figure 3). For (1), let $\mathcal{V} = (V_1, V_2, \ldots, V_T)$ be a sequence of sets of labels corresponding to $\mathcal{Z}$ such that $V_t = \{v_{t,1}, v_{t,2}, \ldots, v_{t,K}\}$ and $v_{t,k}$ is the label at time step $t$ for the entity indexed by $k$. For (2), let $W = (w_1, w_2, \ldots, w_T)$ be a sequence of labels corresponding to $\mathcal{Z}$ where $w_t$ is the label at time step $t$. The goal is then to learn a function $f$ that maps a set of entities and their time-dependent feature vectors $\mathcal{Z}$ to a probability distribution over either (1) the entities' time-dependent labels $\mathcal{V}$ or (2) the sequence of labels $W$.

## 2.2 Multi-agent spatiotemporal modeling

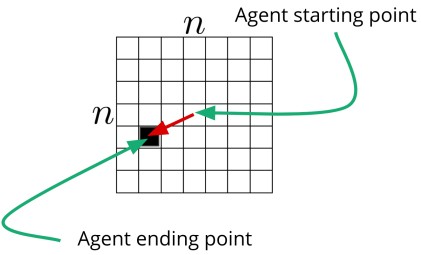

In the MASM setting, $P$ is a set of $K$ different agents and $C_t = \{(x_{t,1}, y_{t,1}), (x_{t,2}, y_{t,2}), \ldots, (x_{t,K}, y_{t,K})\}$ is an unordered set of $K$ coordinate pairs such that $(x_{t,k}, y_{t,k})$ are the coordinates for agent $p_k$ at time step $t$. The ordered sequence of sets of coordinates $\mathcal{C} = (C_1, C_2, \ldots, C_T)$, together with $P$, thus defines the trajectories for the $K$ agents over $T$ time steps. We then define $z_{t,k}$ as: $z_{t,k} = g([e(p_k), x_{t,k}, y_{t,k}, h_{t,k}])$, where $g$ is a multilayer perceptron (MLP), $e$ is an agent embedding layer, and $h_{t,k}$ is a vector of optional contextual features for agent $p_k$ at time step $t$. The trajectory for agent $p_k$ at time step $t$ is defined as $(x_{t+1,k} - x_{t,k}, y_{t+1,k} - y_{t,k})$. Similar to Zheng et al. [10], to fully capture the multimodal nature of the trajectory distributions, we binned the 2D Euclidean space

Figure 2: An example of a binned trajectory. The agent's starting position is at the center of the grid, and the cell containing the agent's ending position is used as the label (of which there are $n^2$ possibilities).

into an $n \times n$ grid (Figure 2) and treated the problem as a classification task. Therefore, $\mathcal{Z}$ has a corresponding sequence of sets of trajectory labels (i.e., $v_{t,k} = \text{Bin}(\Delta x_{t,k}, \Delta y_{t,k})$, so $v_{t,k}$ is an integer from one to $n^2$), and the loss for each sample in **Task P** is: $\mathcal{L} = \sum_{t=1}^{T} \sum_{k=1}^{K} -\ln(f(\mathcal{Z})_{t,k}[v_{t,k}])$, where $f(\mathcal{Z})_{t,k}[v_{t,k}]$ is the probability assigned to the trajectory label for agent $p_k$ at time step $t$ by $f$; i.e., the loss is the NLL of the data according to the model.

For **Task B**, the loss for each sample is: $\mathcal{L} = \sum_{t=1}^{T} -\ln(f(\mathcal{Z})_t[w_t])$, where $f(\mathcal{Z})_t[w_t]$ is the probability assigned to the trajectory label for the ball at time step $t$ by $f$, and the labels correspond to a binned 3D Euclidean space (i.e., $w_t = \text{Bin}(\Delta x_t, \Delta y_t, \Delta z_t)$, so $w_t$ is an integer from one to $n^3$).

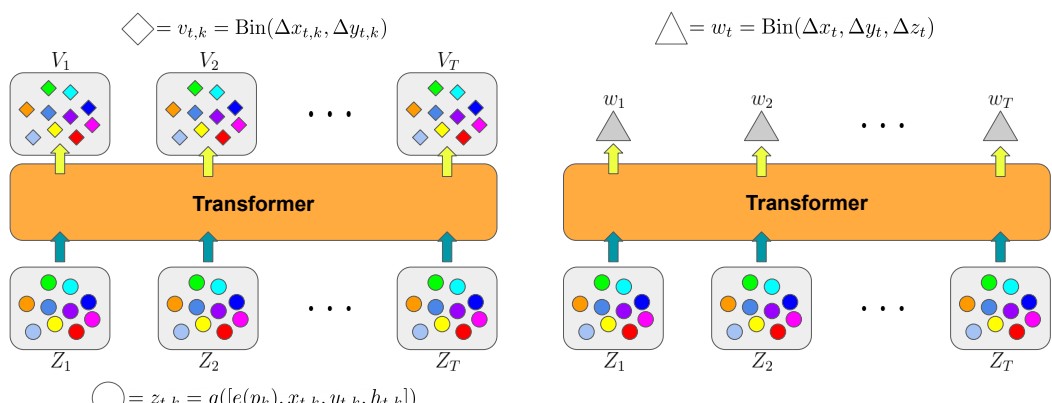

Figure 3: An overview of our multi-entity Transformer, `baller2vec`. Each time step $t$ consists of an *unordered* set $Z_t$ of entity feature vectors (colored circles) as the input, with either (**left**) a corresponding set $V_t$ of entity labels (colored diamonds) or (**right**) a single label $w_t$ (gray triangle) as the target. Matching colored circles/diamonds across time steps correspond to the same entity. In our experiments, each entity feature vector $z_{t,k}$ is produced by an MLP $g$ that takes a player's identity embedding $e(p_k)$, raw court coordinates $(x_{t,k}, y_{t,k})$, and a binary variable indicating the player's frontcourt $h_{t,k}$ as input. Each entity label $v_{t,k}$ is an integer indexing the trajectory bin derived from the player's raw trajectory, while each $w_t$ is an integer indexing the ball's trajectory bin.

## 2.3 The multi-entity Transformer

We now describe our multi-entity Transformer, `baller2vec` (Figure 3). For NLP tasks, the Transformer self-attention mask $M$ takes the form of a $T \times T$ matrix (Figure 4) where $T$ is the length of the sequence. The element at $M_{t_1,t_2}$ thus indicates whether or not the model can "look" at time step $t_2$ when processing time step $t_1$. Here, we generalize the standard Transformer to the multi-entity setting by employing a $T \times K \times T \times K$ mask *tensor* where element $M_{t_1,k_1,t_2,k_2}$ indicates whether or not the model can "look" at agent $p_{k_2}$ at time step $t_2$ when processing agent $p_{k_1}$ at time step $t_1$. Here, we mask all elements where $t_2 > t_1$ and leave all remaining elements unmasked, i.e., `baller2vec` is a "causal" model.

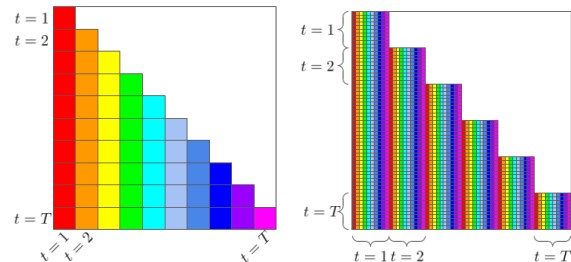

Figure 4: **Left**: the standard self-attention mask matrix $M$. The element at $M_{t_1,t_2}$ indicates whether or not the model can "look" at time step $t_2$ when processing time step $t_1$. **Right**: the matrix form of our multi-entity self-attention mask tensor. In tensor form, element $M_{t_1,k_1,t_2,k_2}$ indicates whether or not the model can "look" at agent $p_{k_2}$ at time step $t_2$ when processing agent $p_{k_1}$ at time step $t_1$. In matrix form, this corresponds to element $M_{t_1 K+k_1, t_2 K+k_2}$ when using zero-based indexing. The $M$ shown here is for a static, fully connected graph, but other, potentially evolving network structures can be encoded in the attention mask tensor.

In practice, to be compatible with Transformer implementations in major deep learning libraries, we reshape $M$ into a $TK \times TK$ matrix (Figure 4), and the input to the Transformer is a matrix with shape $TK \times F$ where $F$ is the dimension of each $z_{t,k}$. Irie et al. [11] observed that positional encoding [5] is not only unnecessary, but detrimental for Transformers that use a causal attention mask, so we do not use positional encoding with `baller2vec`. The remaining computations are identical to the standard Transformer (see code).

## 3 Experiments

### 3.1 Dataset

We trained `baller2vec` on a publicly available dataset of player and ball trajectories recorded from 631 National Basketball Association (NBA) games from the 2015-2016 season.[2] All 30 NBA teams and 450 different players were represented. Because transition sequences are a strategically important part of basketball, unlike prior work, e.g., Felsen et al. [1], Yeh et al. [3], Zhan et al. [2], we did not terminate sequences on a change of possession, nor did we constrain ourselves to a fixed subset of sequences. Instead, each training sample was generated on the fly by first randomly sampling a game, and then randomly sampling a starting time from that game. The following four seconds of data were downsampled to 5 Hz from the original 25 Hz and used as the input.

Because we did not terminate sequences on a change of possession, we could not normalize the direction of the court as was done in prior work [1, 3, 2]. Instead, for each sampled sequence, we randomly (with a probability of 0.5) rotated the court 180° (because the court's direction is arbitrary), doubling the size of the dataset. We used a training/validation/test split of 569/30/32 games, respectively (i.e., 5% of the games were used for testing, and 5% of the remaining 95% of games were used for validation). As a result, we had access to ~82 million different (albeit overlapping) training sequences (569 games × 4 periods per game × 12 minutes per period × 60 seconds per minute × 25 Hz × 2 rotations), ~800x the number of sequences used in prior work. For both the validation and test sets, ~1,000 different, *non-overlapping* sequences were selected for evaluation by dividing each game into $\lceil \frac{1,000}{N} \rceil$ non-overlapping chunks (where $N$ is the number of games), and using the starting four seconds from each chunk as the evaluation sequence.

### 3.2 Model

We trained separate models for **Task P** and **Task B**. For all experiments, we used a single Transformer architecture that was nearly identical to the original model described in Vaswani et al. [5], with $d_{\text{model}} = 512$ (the dimension of the input and output of each Transformer layer), eight attention heads,

---

[2] `https://github.com/linouk23/NBA-Player-Movements`

$d_{\text{ff}} = 2048$ (the dimension of the inner feedforward layers), and six layers, although we did not use dropout. For *both* **Task P** *and* **Task B**, the players *and* the ball were included in the input, and both the players and the ball were embedded to 20-dimensional vectors. The input features for each player consisted of his identity, his $(x, y)$ coordinates on the court at each time step in the sequence, and a binary variable indicating the side of his frontcourt (i.e., the direction of his team's hoop).[3] The input features for the ball were its $(x, y, z)$ coordinates at each time step.

The input features for the players and the ball were processed by separate, three-layer MLPs before being fed into the Transformer. Each MLP had 128, 256, and 512 nodes in its three layers, respectively, and a ReLU nonlinearity following each of the first two layers. For classification, a single linear layer was applied to the Transformer output followed by a softmax. For players, we binned an 11 ft $\times$ 11 ft 2D Euclidean trajectory space into an $11 \times 11$ grid of 1 ft $\times$ 1 ft squares for a total of 121 player trajectory labels. Similarly, for the ball, we binned a 19 ft $\times$ 19 ft $\times$ 19 ft 3D Euclidean trajectory space into a $19 \times 19 \times 19$ grid of 1 ft $\times$ 1 ft $\times$ 1 ft cubes for a total of 6,859 ball trajectory labels.

We used the Adam optimizer [12] with an initial learning rate of $10^{-6}$, $\beta_1 = 0.9$, $\beta_2 = 0.999$, and $\epsilon = 10^{-9}$ to update the model's parameters, of which there were $\sim$19/23 million for **Task P/Task B**, respectively. The learning rate was reduced to $10^{-7}$ after 20 consecutive epochs of the validation loss not improving. Models were implemented in PyTorch and trained on a single NVIDIA GTX 1080 Ti GPU for seven days ($\sim$650 epochs) where each epoch consisted of 20,000 training samples, and the validation set was used for early stopping.

## 3.3 Baselines

As our naive baseline, we used the marginal distribution of the trajectory bins from the training set for all predictions. For our strong baseline, we implemented a `baller2vec`-like graph recurrent neural network (GRNN) and trained it on **Task P** (code is available in the `baller2vec` repository).[4] Specifically, at each time step, the player and ball inputs were first processed using MLPs as in `baller2vec`, and these inputs were then fed into a graph neural network (GNN) similar to Yeh et al. [3]. The node and edge functions of the GNN were each a Transformer-like feedforward network (TFF), i.e., $\text{TFF}(x) = \text{LN}(x + W_2\text{ReLU}(W_1x + b_1) + b_2)$, where LN is Layer Normalization [13], $W_1$ and $W_2$ are weight matrices, $b_1$ and $b_2$

Table 1: The perplexity per trajectory bin on the test set when using `baller2vec` vs. the marginal distribution of the trajectory bins in the training set ("Train") for all predictions. `baller2vec` considerably reduces the uncertainty over the trajectory bins.

|  | `baller2vec` | Train |
|---|---|---|
| **Task P** | 1.64 | 15.72 |
| **Task B** | 13.44 | 316.05 |

are bias vectors, and ReLU is the rectifier activation function. For our RNN, we used a gated recurrent unit (GRU) RNN [14] in which we replaced each of the six weight matrices of the GRU with a TFF. Each TFF had the same dimensions as the Transformer layers used in `baller2vec`. Our GRNN had $\sim$18M parameters, which is comparable to the $\sim$19M in `baller2vec`. We also trained our GRNN for seven days ($\sim$175 epochs).

## 3.4 Ablation studies

To assess the impacts of the multi-entity design and player embeddings of `baller2vec` on model performance, we trained three variations of our **Task P** model using: (1) one player in the input without player identity, (2) all 10 players in the input without player identity, and (3) all 10 players in the input with player identity. In experiments where player identity was not used, a single generic player embedding was used in place of the player identity embeddings. We also trained two variations of our **Task B** model: one with player identity and one without. Lastly, to determine the extent to which `baller2vec` uses historical information in its predictions, we compared the performance of our best **Task P** model on the full sequence test set with its performance on the test set when *only predicting the trajectories for the first frame* (i.e., we applied the *same* model to only the first frames of the test set).

---

[3]We did not include team identity as an input variable because teams are collections of players and a coach, and coaches did not vary in the dataset because we only had access to half of one season of data; however, with additional seasons of data, we would include the coach as an input variable.

[4]We chose to implement our own strong baseline because `baller2vec` has far more parameters than models from prior work (e.g., $\sim$70x Felsen et al. [1]).

## 4 Results

### 4.1 `baller2vec` is an effective learning algorithm for multi-agent spatiotemporal modeling.

The average NLL on the test set for our best **Task P** model was 0.492, while the average NLL for our best **Task B** model was 2.598. In NLP, model performance is often expressed in terms of the perplexity per word, which, intuitively, is the number of faces on a fair die that has the same amount of uncertainty as the model per word (i.e., a uniform distribution over $M$ labels has a perplexity of $M$, so a model with a per word perplexity of six has the same average uncertainty as rolling a fair six-sided die). In our case, we con-

Table 2: The average NLL (lower is better) on the **Task P** test set and seconds per training epoch (SPE) for `baller2vec` (b2v) and our GRNN. `baller2vec` trains ∼3.8 times faster per epoch compared to our GRNN, and `baller2vec` outperformed our GRNN by 10.5% when given the same amount of training time. Even when only allowed to train for half ("0.5x") and a quarter ("0.25x") as long as our GRNN, `baller2vec` outperformed our GRNN by 9.1% and 1.5%, respectively..

|  | b2v | b2v (0.5x) | b2v (0.25x) | GRNN |
|---|---|---|---|---|
| NLL | 0.492 | 0.499 | 0.541 | 0.549 |
| SPE | ∼900 | ∼900 | ∼900 | ∼3,400 |

sider the perplexity per trajectory bin, defined as: $PP = e^{\frac{1}{NTK} \sum_{n=1}^{N} \sum_{t=1}^{T} \sum_{k=1}^{K} -\ln(p(v_{n,t,k}))}$, where $N$ is the number of sequences. Our best **Task P** model achieved a $PP$ of 1.64, i.e., `baller2vec` was, on average, as uncertain as rolling a 1.64-sided fair die (better than a coin flip) when predicting player trajectory bins (Table 1). For comparison, when using the distribution of the player trajectory bins in the training set as the predicted probabilities, the $PP$ on the test set was 15.72. Our best **Task B** model achieved a $PP$ of 13.44 (compared to 316.05 when using the training set distribution).

Compared to our GRNN, `baller2vec` was ∼3.8 times faster and had a 10.5% lower average NLL when given an equal amount of training time (Table 2). Even when only given half as much training time as our GRNN, `baller2vec` had a 9.1% lower average NLL.

### 4.2 `baller2vec` uses information about all players on the court through time, in addition to player identity, to model spatiotemporal dynamics.

Results for our ablation experiments can be seen in Table 3. Including all 10 players in the input dramatically improved the performance of our **Task P** model by 18.0% vs. only including a single player. Including player identity improved the model's performance a further 4.4%. This stands in contrast to Felsen et al. [1] where the inclusion of player identity led to slightly *worse* model performance; a counterintuitive result given the range of skills among NBA players, but possibly a side effect of their role-alignment procedure. Additionally, when replacing the players in each test set sequence with random players, the performance of our best **Task P** model deteriorated by 6.2% from 0.492 to 0.522. Interestingly, including player identity only improved our **Task B** model's performance by 2.7%. Lastly, our best **Task P** model's performance on the full sequence test set

Table 3: The average NLL on the test set for each of the models in our ablation experiments (lower is better). For **Task P**, using all 10 players improved model performance by 18.0%, while using player identity improved model performance by an additional 4.4%. For **Task B**, using player identity improved model performance by 2.7%. 1/10 indicates whether one or 10 players were used as input, respectively, while I/NI indicates whether or not player identity was used, respectively.

| Task | 1-NI | 10-NI | 10-I |
|---|---|---|---|
| **Task P** | 0.628 | 0.515 | 0.492 |
| **Task B** | N/A | 2.670 | 2.598 |

(0.492) was 70.6% better than its performance on the single frame test set (1.67), i.e., `baller2vec` is clearly using historical information to model the spatiotemporal dynamics of basketball.

### 4.3 `baller2vec`'s learned player embeddings encode individual attributes.

Neural language models are widely known for their ability to encode semantic relationships between words and phrases as geometric relationships between embeddings—see, e.g., Mikolov et al. [16, 17], Le and Mikolov [18], Sutskever et al. [19]. Alcorn [20] observed a similar phenomenon in a baseball setting, where batters and pitchers with similar skills were found next to each other in the

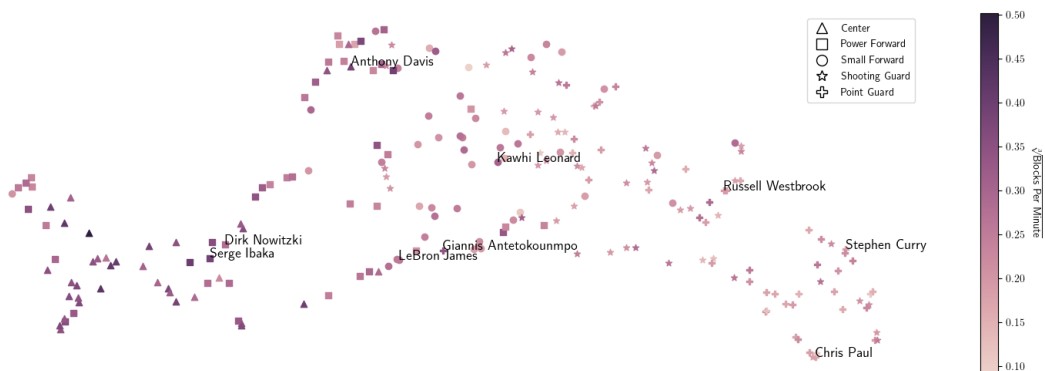

Figure 5: As can be seen in this 2D UMAP of the player embeddings, by exclusively learning to predict the trajectory of the ball, `baller2vec` was able to infer idiosyncratic player attributes. The left-hand side of the plot contains tall post players ($\triangle$, $\square$), e.g., Serge Ibaka, while the right-hand side of the plot contains shorter shooting guards ($\star$) and point guards (+), e.g., Stephen Curry. The connecting transition region contains forwards ($\square$, $\bigcirc$) and other "hybrid" players, i.e., individuals possessing both guard and post skills, e.g., LeBron James. Further, players with similar defensive abilities, measured here by the cube root of the players' blocks per minute in the 2015-2016 season [15], cluster together.

embedding space learned by a neural network trained to predict the outcome of an at-bat. A 2D UMAP [21] of the player embeddings learned by `baller2vec` for **Task B** can be seen in Figure 5. Like (`batter|pitcher`)`2vec` [20], `baller2vec` seems to encode skills and physical attributes in its player embeddings.

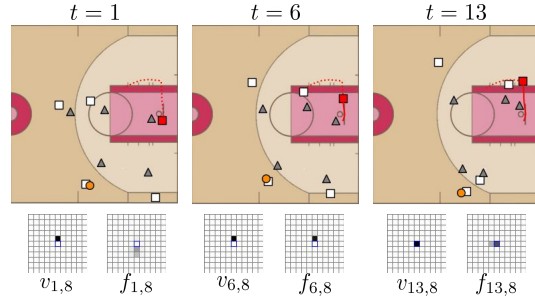

Figure 6: `baller2vec`'s trajectory predicted trajectory bin distributions are affected by both the historical and current context. At $t = 1$, `baller2vec` is fairly uncertain about the target player's ($\blacktriangle$; $k = 8$) trajectory (left grid and dotted red line; the blue-bordered center cell is the "stationary" trajectory), with most of the probability mass divided between trajectories moving towards the ball handler's sideline (right grid; black = 1.0; white = 0.0). After observing a portion of the sequence ($t = 6$), `baller2vec` becomes very certain about the target player's trajectory ($f_{6,8}$), but when the player reaches a decision point ($t = 13$), `baller2vec` becomes split between trajectories (staying still or moving towards the top of the key). Additional examples can be found in Figure S1. ● = ball, $\square$ = offense, $\blacktriangle$ = defense, and $f_{t,k} = f(\mathcal{Z})_{t,k}$.

Querying the nearest neighbors for individual players reveals further insights about the `baller2vec` embeddings. For example, the nearest neighbor for Russell Westbrook, an extremely athletic 6'3" point guard, is Derrick Rose, a 6'2" point guard also known for his athleticism. Amusingly, the nearest neighbor for Pau Gasol, a 7'1" center with a respectable shooting range, is his younger brother Marc Gasol, a 6'11" center, also with a respectable shooting range.

### 4.4 `baller2vec`'s predicted trajectory bin distributions depend on both the historical and current context.

Because `baller2vec` *explicitly* models the distribution of the player trajectories (unlike variational methods), we can easily visualize how its predicted trajectory bin distributions shift in different situations. As can be seen in Figure 6, `baller2vec`'s predicted trajectory bin distributions depend on both the historical and current context. When provided with limited historical information, `baller2vec` tends to be less certain about where the players might go. `baller2vec` also tends to be more certain when predicting trajectory bins at "easy" moments (e.g., a player moving into open space) vs. "hard" moments (e.g., an offensive player choosing which direction to move around a defender).

### 4.5 Attention heads in `baller2vec` appear to perform basketball-relevant functions.

One intriguing property of the attention mechanism [22–25] is how, when visualized, the attention weights often seem to reveal how a model is "thinking". For example, Vaswani et al. [5] discovered examples of attention heads in their Transformer that appear to be performing various language understanding subtasks, such as anaphora resolution. As can be seen in Figure 7, some of the attention heads in `baller2vec` seem to be performing basketball understanding subtasks, such as keeping track of the ball handler's teammates, and anticipating who the ball handler will pass to, which, intuitively, help with our task of predicting the ball's trajectory.

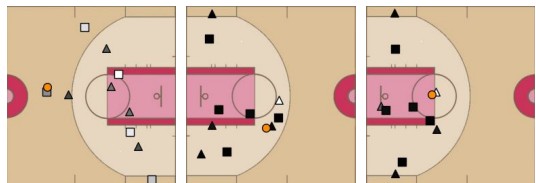

Figure 7: The attention outputs from `baller2vec` suggest it learned basketball-relevant functions. **Left**: attention head 2-7 (layer-head) appears to focus on teammates of the ball handler (■●). **Middle and right**: attention head 6-2 seems to predict (middle; △) who the ball handler will be in a future frame (right). Players are shaded according to the *sum* of the attention weights assigned to the players *through time* with reference to the ball in the current frame (recall that each player occurs multiple times in the input). Higher attention weights are lighter. For both of these attention heads, the sum of the attention weights assigned to the ball through time was small (0.01 for both the left and middle frames where the maximum is 1.00). Additional examples can be found in Figures S2 and S3.

## 5 Related Work

### 5.1 Trajectory modeling in sports

There is a rich literature on MASM, particularly in the context of sports, e.g., Kim et al. [26], Zheng et al. [10], Le et al. [27, 28], Qi et al. [29], Zhan et al. [30]. Most relevant to our work is Yeh et al. [3], who used a variational recurrent neural network combined with a graph neural network to forecast trajectories in a multi-agent setting. Like their approach, our model is permutation equivariant with regard to the ordering of the agents; however, we use a multi-head attention mechanism to achieve this permutation equivariance while the permutation equivariance in Yeh et al. [3] is provided by the graph neural network. Specifically, Yeh et al. [3] define: $v \rightarrow e : \mathbf{e}_{i,j} = f_e([\mathbf{v}_i, \mathbf{v}_j, \mathbf{t}_{i,j}])$ and $e \rightarrow v : \mathbf{o}_i = f_v(\sum_{j \in \mathcal{N}_i}[\mathbf{e}_{i,j}, \mathbf{t}_i])$, where $\mathbf{v}_i$ is the initial state of agent $i$, $\mathbf{t}_{i,j}$ is an embedding for the edge between agents $i$ and $j$, $\mathbf{e}_{i,j}$ is the representation for edge $(i, j)$, $\mathcal{N}_i$ is the neighborhood for agent $i$, $\mathbf{t}_i$ is a node embedding for agent $i$, $\mathbf{o}_i$ is the output state for agent $i$, and $f_e$ and $f_v$ are deep neural networks.

Assuming *each individual player* is a different "type" in $f_e$ (i.e., attempting to maximize the level of personalization) would require $450^2 = 202{,}500$ (i.e., $B^2$) different $t_{i,j}$ edge embeddings, many of which would never be used during training and thus inevitably lead to poor out-of-sample performance. Reducing the number of type embeddings requires making assumptions about the nature of the relationships between nodes. By using a multi-head attention mechanism, `baller2vec` learns to integrate information about different agents in a highly flexible manner that is both agent and time-dependent, and can generalize to unseen agent combinations. The attention heads in `baller2vec` are somewhat analogous to edge types, but, importantly, they do not require a priori knowledge about the relationships between the players.

Additionally, unlike recent works that use variational methods to train their generative models [3, 1, 2], we translate the multi-agent trajectory modeling problem into a classification task, which allows us to train our model by strictly maximizing the likelihood of the data. As a result, we do not make any assumptions about the distributions of the trajectories nor do we need to set any priors over latent variables. Zheng et al. [10] also predicted binned trajectories, but they used a recurrent convolutional neural network to predict the trajectory for a single player trajectory at a time at each time step.

### 5.2 Transformers for multi-agent spatiotemporal modeling

Giuliari et al. [31] used a Transformer to forecast the trajectories of *individual* pedestrians, i.e., the model does not consider interactions between individuals. Yu et al. [9] used *separate* temporal and spatial Transformers to forecast the trajectories of multiple, interacting pedestrians. Specifically, the temporal Transformer processes the coordinates of each pedestrian *independently* (i.e., it does not model interactions), while the spatial Transformer, which is inspired by Graph Attention Networks

[8], processes the pedestrians *independently at each time step*. Sanford et al. [32] used a Transformer to classify on-the-ball events from sequences in soccer games; however, only the coordinates of the $K$-nearest players to the ball were included in the input (along with the ball's coordinates). Further, the *order* of the included players was based on their average distance from the ball for a given temporal window, which can lead to specific players changing position in the input between temporal windows. As far as we are aware, baller2vec is the **first** Transformer capable of processing all agents *simultaneously across time* without imposing an order on the agents.

# 6   Limitations

While baller2vec does not have a mechanism for handling unseen players, a number of different solutions exist depending on the data available. For example, similar to what was proposed in Alcorn [20], a model could be trained to map a vector of (e.g., NCAA) statistics and physical measurements to baller2vec embeddings. Alternatively, if tracking data is available for the other league, a single baller2vec model could be jointly trained on all the data.

At least two different factors may explain why including player identity as an input to baller2vec only led to relatively small performance improvements. First, both player and ball trajectories are fairly generic—players tend to move into open space, defenders tend to move towards their man or the ball, point guards tend to pass to their teammates, and so on. Further, the location of a player on the court is often indicative of their position, and players playing the same position tend to have similar skills and physical attributes. As a result, we might expect baller2vec to be able to make reasonable guesses about a player's/ball's trajectory just given the location of the players and the ball on the court.

Second, baller2vec may be able to *infer* the identities of the players directly from the spatiotemporal data. Unlike (batter|pitcher)2vec [20], which was trained on several seasons of Major League Baseball data, baller2vec only had access to one half of one season's worth of NBA data for training. As a result, player identity may be entangled with season-specific factors (e.g., certain rosters or coaches) that are actually exogenous to the player's intrinsic qualities, i.e., baller2vec may be overfitting to the season. To provide an example, the Golden State Warriors ran a very specific kind of offense in the 2015-2016 season—breaking the previous record for most three-pointers made in the regular season by 15.4%—and many basketball fans could probably recognize them from a bird's eye view (i.e., without access to any identifying information). Given additional seasons of data, baller2vec would no longer be able to exploit the implicit identifying information contained in static lineups and coaching strategies, so including player identity in the input would likely be more beneficial in that case.

# 7   Conclusion

In this paper, we introduced baller2vec, a generalization of the standard Transformer that can model sequential data consisting of multiple, unordered entities at each time step. As an architecture that both is computationally efficient and has powerful representational capabilities, we believe baller2vec represents an exciting new direction for MASM. As discussed in Section 6, training baller2vec on more training data may allow the model to more accurately factor players away from season-specific patterns. With additional data, more contextual information about agents (e.g., a player's age, injury history, or minutes played in the game) and the game (e.g., the time left in the period or the score difference) could be included as input, which might allow baller2vec to learn an even more complete model of the game of basketball. Although we only experimented with static, fully connected graphs here, baller2vec can easily be applied to more complex inputs—for example, a sequence of graphs with changing nodes and edges—by adapting the self-attention mask tensor as appropriate. Lastly, as a generative model (see Alcorn and Nguyen [33] for a full derivation), baller2vec could be used for counterfactual simulations (e.g., assessing the impact of different rosters), or combined with a controller to discover optimal play designs through reinforcement learning.

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
