# OpenReview forum: "baller2vec: A Multi-Entity Transformer For Multi-Agent Spatiotemporal Modeling"
_NeurIPS.cc/2021/Conference — NeurIPS 2021 Submitted_

### Official Review · Reviewer_7Vsi · 2021-07-14

**Rating:** 2
**Confidence:** 5

**Summary:**

The authors introduce a transformer-based approach to model ball and player-trajectories in basketball.  Advantages of this approach include the ability to perform multi-agent trajectory prediction without requiring a strict ordering.  The approach itself, is not novel- it is a simple application of transformer architecture.  Additionally, this approach is only demonstrated on a single dataset in a specific setting.  Furthermore, model performance is only reported in terms of perplexity, not RMSE on predicted paths, and this prevents fair comparison to past approaches.

**Limitations And Societal Impact:**

The authors mention that this approach does not have a mechanism for handling unseen players.  This is a significant limitation as other past approaches have been able to address such an issue.  This is further an issue because unseen players is a common issue in sport-player tracking.  It is an even more serious issue in more unconstrained settings such as pedestrian tracking.  The authors mention possible ways to address this, but without exploring these and demonstrating this severe limitation can be overcome, the usefulness of this method is severely limited.

Similarly, limitations and impact were discussed only in the context of sports.  Presumably this method should extend to multi-agent modeling in a variety of settings, many of which have surveillance-related applications.  Limitations and impact in such other domains is noticeably absent.

**Main Review:**

This work is largely an application of transformer architecture to a team-sports setting.  There is minimal originality or innovation on the algorithm side.

This paper is a resubmission of the work to ICML 2021.  The only significant change is the addition of the "naive baselines" in Section 3.3.  While additional baselines are helpful, issues remain.  First, this is not a "naive baseline"- this is using a GRNN similar to Yeh et al. with additional transformer features.  Because this is not a standard approach, it cannot be judged if this is a poor architecture decision (or trained poorly) or how it compares to the proposed method.  Comparing to Yeh et al directly would be a better approach.  Additionally, a true "naive baseline" would use something like constant velocity prediction or dictionary lookup.

In addition to the baseline not being properly employed and tested, there is the remaining issue that simply adding this one section does not make this a sufficiently different or improved submission.  This paper was rejected at ICML, and there has not been sufficient improvement to warrant acceptance here.

Model performance is given in terms of perplexity, which is fine, but prevents comparison to other approaches.   Previous work in this area including Yeh et al, Felsen et al, etc. all report RMSE on the final trajectory.  It is understood that this approach allows for multiple potential outcomes, but then calculate RMSE on the top-k paths.  The unwillingness of the authors to report such a metric suggests that this approach does not compare favorably to past approaches.

Overall the comparison to past approaches (both in terms of metrics provided as well as which methods are compared) is severely lacking and a major shortcoming of the work.  Without comparing to other approaches (dictionary lookup, Yeh et al, Felsen et al, Social LSTM, Social GAN, etc.), I don't believe this paper can be accepted.

Figure 5 and the surrounding analysis is probably not well suited for a NeurIPS publication.  The embeddings shown there are not substantially different than those obtained by other methods.  Furthermore, there is no way to demonstrate they are "better".  Such analysis is better suited for a sports-focused venue/publication.

Sports is a useful setting for studying multi-agent motion.  However, given the assertion of this work is that this method is broadly useful for multi-agent trajectory forecasting.  That has not been demonstrated as only a single dataset has been used.  To demonstrate its usefulness, other multi-agent datasets should be explored.

The claim that GNNs limit personalization (line 32) is false.  Player embeddings or other approaches are easily incorporated into such frameworks.

In general the figures are not as useful as they could be- it is difficult to comprehend what the output trajectories look like.  Figures 3 and 4 are not particularly useful or illuminating either.



**Time Spent Reviewing:**

2

---

> ### Author Response · Authors · 2021-08-06
> **Clarifying Our Contributions**
>
> **There is minimal originality or innovation on the algorithm side.**
>
> While we recognize that `baller2vec` is not a major technical innovation in terms of model components, we believe our reinterpretation of the attention mask matrix as a reshaped tensor is valuable because it allows the standard Transformer architecture to be applied to a broader class of problems (similar to ViT (Dosovitskiy et al., 2021)—an ICLR 2021 paper [1]). The fact that, in the 3.5 years between when the original Transformer manuscript and `baller2vec` were published, no other researchers had attempted to use a Transformer for multi-agent spatiotemporal modeling in this way suggests this approach was not a priori obvious. Further, [like Yoshua Bengio](https://www.facebook.com/100002848499232/posts/2805055846266004/), we believe that the application of existing technologies to challenging, real-world problems (MASM in our paper) contributes valuable knowledge to the community.
>
> **First, this is not a "naive baseline"- this is using a GRNN similar to Yeh et al. with additional transformer features.**
>
> We explicitly call the GRNN “our strong baseline” (Line 191).
>
> **Comparing to Yeh et al directly would be a better approach.**
>
> Yes! We made every attempt to compare with Yeh et al. (2019), including implementing our own GRNN version of their approach. However, Yeh et al. (2019) did not include the hyperparameters of their architecture in their paper, did not publicly release their code (as we pointed out in our ICML rebuttal), and did not reply to an email request for code/model hyperparameters, making direct comparison to their approach impossible. Unlike Yeh et al. (2019) and many other papers that use sports datasets, the code for our entire pipeline, from data generation to model training, is public, which, as we elaborate below, is an important contribution of our work to the community.
>
> **In addition to the baseline not being properly employed and tested, there is the remaining issue that simply adding this one section does not make this a sufficiently different or improved submission.**
>
> The primary shared concern of the ICML reviewers was the lack of a comparison to a strong baseline, which we added for our NeurIPS submission. Can you justify your assertion that our baseline was “not properly employed and tested”?
>
> **The unwillingness of the authors to report such a metric suggests that this approach does not compare favorably to past approaches.**
>
> We compared `baller2vec` to a strong baseline inspired by prior work using our chosen performance metric. RMSE is not a more valid performance metric than the NLL metric we reported in the manuscript. We note that the other NeurIPS reviewers did not express similar concerns about our chosen performance metric, which was also the case with the other ICML reviewers. We cannot directly compare to the reported results in previous papers because we were using a different (much larger) training dataset and a different task (“language modeling” the trajectories for multiple agents). In fact, we feel the lack of standardized benchmarks for sports datasets, combined with the general lack of open source code, significantly complicates comparing different models and has hindered the pace of research in this domain. As alluded to above, one of the important contributions of our research is including code to reproduce our full pipeline, which will allow other researchers to easily compare directly to our work.
>
> **That has not been demonstrated as only a single dataset has been used. To demonstrate its usefulness, other multi-agent datasets should be explored.**
>
> Unfortunately, the commonly used pedestrian datasets (i.e., ETH, HOTEL, UNIV, ZARA1, and ZARA2) are *extremely* small, which makes them insufficient for evaluating an architecture that is known to thrive on large datasets (researchers using Transformers for language modeling are not expected to evaluate their architectures on datasets containing thousands of sentences). Specifically, whereas our NBA dataset has ~82 million different time steps in the training set, the total number of time steps for all of the previously mentioned pedestrian datasets combined is only 6,982. Likewise, the [H3D traffic dataset](https://usa.honda-ri.com/H3D) (Patil et al., 2020 [2]) only contains 27,721 time steps.
>
> We also note that Social-BiGAT (Kosaraju et al., 2019), a NeurIPS 2019 paper [3], was exclusively evaluated on the previously mentioned pedestrian datasets, and Zhan et al. (2019), an ICLR 2019 paper [4], exclusively evaluated their model on an NBA dataset, so there is precedent in the community to evaluate multi-agent architectures in a single domain. However, we look forward to the opportunity to evaluate `baller2vec` on other large, multi-agent datasets when they become available.
>
> **The claim that GNNs limit personalization (line 32) is false. Player embeddings or other approaches are easily incorporated into such frameworks.**
>
> As we clarified in our ICML rebuttal, the personalization limitations of Yeh et al. (2019) refers specifically to their edge types (as we explicitly explain starting at Line 334).
>
> **Figures 3 and 4 are not particularly useful or illuminating either.**
>
> Can you explain why these figures are not useful or illuminating? Reviewer GDru stated, “Fig. 2, 3, 4 were helpful in understanding the method.”.
>
> **The authors mention that this approach does not have a mechanism for handling unseen players. This is a significant limitation as other past approaches have been able to address such an issue. This is further an issue because unseen players is a common issue in sport-player tracking. It is an even more serious issue in more unconstrained settings such as pedestrian tracking.**
>
> As we demonstrated in our ablation experiments, `baller2vec` does not require identities to perform well (see Table 3), so we do not feel there is any reason to believe the model would struggle with a pedestrian dataset. For the sports setting where new players can join a league, we proposed a number of practical solutions for handling unseen players depending on the available data. There are also many other potential solutions we did not discuss in the paper, such as randomly replacing a player’s identity embedding with a position embedding during training.
>
> **Presumably this method should extend to multi-agent modeling in a variety of settings, many of which have surveillance-related applications. Limitations and impact in such other domains is noticeably absent.**
>
> As we explained in our checklist response (Line 500), our architecture does not introduce any *new* ethical challenges. Every multi-agent trajectory model has similar ethical issues, so we felt that it would be an inefficient use of space to spell those out, particularly since NeurIPS did not require a broader impacts section this year.
>
> **References**
>
> [1] Alexey Dosovitskiy, Lucas Beyer, Alexander Kolesnikov, Dirk Weissenborn, Xiaohua Zhai, Thomas Unterthiner, Mostafa Dehghani, Matthias Minderer, Georg Heigold, Sylvain Gelly, Jakob Uszkoreit, and Neil Houlsby. An image is worth 16x16 words: Transformers for image recognition at scale. In International Conference on Learning Representations, 2021.
>
> [2] Abhishek Patil, Srikanth Malla, Haiming Gang, and Yi-Ting Chen. The h3d dataset for full-surround 3d multi-object detection and tracking in crowded urban scenes. In International Conference on Robotics and Automation, 2019.
>
> [3] Vineet Kosaraju, Amir Sadeghian, Roberto Martín-Martín, Ian Reid, Hamid Rezatofighi, and Silvio Savarese. Social-bigat: Multimodal trajectory forecasting using bicycle-gan and graph attention networks. In Advances in Neural Information Processing Systems, 2019.
>
> [4] Eric Zhan, Stephan Zheng, Yisong Yue, Long Sha, and Patrick Lucey. Generating multi-agent trajectories using programmatic weak supervision. In International Conference on Learning Representations, 2019.

---

### Official Review · Reviewer_GDru · 2021-07-16

**Rating:** 5
**Confidence:** 4

**Summary:**

In this work, a Transformer architecture is proposed which generalizes the typical Transformer model to be able to leverage information across entities and time (previously, models only operated with information from one or the other). It does this with a new block-lower-triangular self-attention mask matrix (actually a tensor, but reshaped to work with existing software frameworks). The method is experimentally verified on a dataset of NBA basketball games and the trajectories of players and the ball.

**Limitations And Societal Impact:**

While the authors have included a limitations section, they have not written a broader impact section.

**Main Review:**

Strengths:
- The work tackles an important problem which can be applied to many domains.

- The authors present a novel multi-entity generalization of the typical Transformer via a joint entity-temporal attention scheme, improving the applicability of Transformer models to domains which have multiple entities interacting across time.

- The paper is written clearly. In particular, Fig. 2, 3, 4 were helpful in understanding the method.

- There are many ablation studies which aid in understanding the sources of performance for the model.

Weaknesses:

- Related Work:
    - There is an earlier work related to graph-based basketball player trajectory forecasting that is worth adding to the related work discussion [1].

- Experiments:
    - Using a 95%/5% trainval-test split seems quite heavy on the training/validation side, a more common split might be 70/30 or 80/20 (the smaller the testing size, the less diversity of scenarios the model is tested on).
    - The implemented model is _huge_. Was it necessary for the player and ball input features (which I understand to be a few numbers encoded to 20-dim vectors, Line 171) to be processed by 3-layer MLPs with hundreds of nodes per layer? Does the ~70x increase in parameters result in significantly better performance? It would be interesting to see a comparison to Felsen et al. [1] or other similar works even if they do have significantly less parameters (runtime is perhaps more of a consideration here rather than training time as in Table 2).
    - While the authors did implement some relevant baselines, there are other transformer models for trajectory forecasting (with open-sourced code) that would serve as good (and arguably important) comparisons. For instance: https://github.com/FGiuliari/Trajectory-Transformer (while it does not model interactions, it is still a worthwhile comparison to show the power of modeling interactions) or some of the other cited methods in Section 5.2.
    - The experiments are only conducted on NBA data. It is difficult to make the claim that baller2vec is an effective learning algorithm for the general problem of MASM without showing its performance on other MASM problems (e.g., more typical pedestrian or traffic motion, such as that found in the nuScenes dataset, for example).

[1] B. Ivanovic, E. Schmerling, K. Leung, and M. Pavone, “Generative Modeling of Multimodal Multi-Human Behavior,” in IEEE/RSJ Int. Conf. on Intelligent Robots & Systems, Madrid, Spain, 2018.

**Time Spent Reviewing:**

4

---

> ### Author Response · Authors · 2021-08-06
> **Regarding Our Model Size and Evaluation Experiments**
>
> Thank you for your review and for highlighting the strengths of our manuscript! We will address your other comments below, and make the below points clearer in the manuscript.
>
> **There is an earlier work related to graph-based basketball player trajectory forecasting that is worth adding to the related work discussion [1].**
>
> Thank you for bringing this reference to our attention. We will cite Ivanovic et al. (2018) in the Related Work section.
>
> **Using a 95%/5% trainval-test split seems quite heavy on the training/validation side, a more common split might be 70/30 or 80/20 (the smaller the testing size, the less diversity of scenarios the model is tested on).**
>
> Because our dataset is ~800x larger than other NBA datasets, e.g., Felsen et al. (2018) [1] and Zhan et al. (2019) [2], using 5% of the games for our test set allows us to use more training data while keeping the test set large enough to be an accurate estimate of the model’s performance. Specifically, our test set consists of ~1,000 different sequences (i.e., ~20,000 time steps) divided among 32 different games, which corresponds to ~30 different sequences per game, which we feel was sufficiently diverse for evaluation purposes. For comparison, the commonly used pedestrian datasets (i.e., ETH, HOTEL, UNIV, ZARA1, and ZARA2) only have 6,982 time steps of data *combined*. We also note that our model did not overfit during training, so the performance of our model on the training set and test set was similar (our best training loss was 0.495 compared to our test loss of 0.492).
>
> **The implemented model is huge. Was it necessary for the player and ball input features (which I understand to be a few numbers encoded to 20-dim vectors, Line 171) to be processed by 3-layer MLPs with hundreds of nodes per layer? Does the ~70x increase in parameters result in significantly better performance? It would be interesting to see a comparison to Felsen et al. [1] or other similar works even if they do have significantly less parameters (runtime is perhaps more of a consideration here rather than training time as in Table 2).**
>
> As stated above, our model did not overfit during training, which suggests our model was underparameterized if anything. Unfortunately, Yeh et al. (2019) did not include the hyperparameters of their architecture in their paper, did not publicly release their code, and did not reply to an email request for code/model hyperparameters, which made direct comparison to their approach impossible. While Felsen et al. (2018) did include their neural network’s hyperparameters in their paper, code for their “role-alignment” procedure is not available. Unlike Yeh et al. (2019), Felsen et al. (2018), and many other papers that use sports datasets, the code for our entire pipeline, from data generation to model training, is public, which is an important contribution of our work to the community. We feel the lack of standardized benchmarks for sports datasets, combined with the general lack of open source code, significantly complicates comparing different models and has hindered the pace of research in this domain.
>
> **While the authors did implement some relevant baselines, there are other transformer models for trajectory forecasting (with open-sourced code) that would serve as good (and arguably important) comparisons. For instance: https://github.com/FGiuliari/Trajectory-Transformer (while it does not model interactions, it is still a worthwhile comparison to show the power of modeling interactions) or some of the other cited methods in Section 5.2.**
>
> Our single player input ablation experiment (introduced at Line 209 with results in Table 3) is effectively the Giuliari et al. (2020) model in that it is a vanilla Transformer applied to a single agent. As shown in Table 3, our 10-player model outperformed our single-player model by 18.0%.
>
> **The experiments are only conducted on NBA data. It is difficult to make the claim that baller2vec is an effective learning algorithm for the general problem of MASM without showing its performance on other MASM problems (e.g., more typical pedestrian or traffic motion, such as that found in the nuScenes dataset, for example).**
>
> Unfortunately, as mentioned above, the commonly used pedestrian datasets are *extremely* small, which makes them insufficient for evaluating an architecture that is known to thrive on large datasets (researchers using Transformers for language modeling are not expected to evaluate their architectures on idiosyncratic datasets containing thousands of sentences). Specifically, whereas our NBA dataset has ~82 million different time steps in the training set, the total number of time steps for all of the previously mentioned pedestrian datasets combined is only 6,982. Likewise, the [H3D traffic dataset](https://usa.honda-ri.com/H3D) (Patil et al., 2020 [3]) only contains 27,721 time steps.
>
> We also note that Social-BiGAT (Kosaraju et al., 2019), a NeurIPS 2019 paper [4], was exclusively evaluated on the previously mentioned pedestrian datasets, and Zhan et al. (2019), an ICLR 2019 paper [2], exclusively evaluated their model on an NBA dataset, so there is precedent in the community to evaluate multi-agent architectures in a single domain. If the primary issue is that our language is too strong, we are happy to adjust our wording. Otherwise, we look forward to the opportunity to evaluate `baller2vec` on other large, multi-agent datasets when they become available.
>
> **References**
>
> [1] Panna Felsen, Patrick Lucey, and Sujoy Ganguly. Where will they go? predicting fine-grained adversarial multi-agent motion using conditional variational autoencoders. In Proceedings of the European Conference on Computer Vision, 2018.
>
> [2] Eric Zhan, Stephan Zheng, Yisong Yue, Long Sha, and Patrick Lucey. Generating multi-agent trajectories using programmatic weak supervision. In International Conference on Learning Representations, 2019.
>
> [3] Abhishek Patil, Srikanth Malla, Haiming Gang, and Yi-Ting Chen. The h3d dataset for full-surround 3d multi-object detection and tracking in crowded urban scenes. In International Conference on Robotics and Automation, 2019.
>
> [4] Vineet Kosaraju, Amir Sadeghian, Roberto Martín-Martín, Ian Reid, Hamid Rezatofighi, and Silvio Savarese. Social-bigat: Multimodal trajectory forecasting using bicycle-gan and graph attention networks. In Advances in Neural Information Processing Systems, 2019.

---

> > ### Comment · Reviewer_GDru · 2021-08-31
> > **Response to Authors**
> >
> > Thank you for your response, it is informative and appreciated.
> >
> > I believe most of my identified issues can be fixed in the paper by writing the above responses in the paper. However, the key outstanding weakness of the work is the lack of evaluation on other (public) datasets. The authors correctly point out that commonly-used pedestrian datasets are very small. This is well-known in the field, however, and is the reason why I mentioned the nuScenes dataset in my initial review. If nuScenes is still too small, the Lyft Level 5 dataset features 1,118 hours of data at 10 Hz (=> 40.2 million different timesteps), which is plenty.
> >
> > Additionally, the authors should be careful with statements like "Paper A & B exclusively evaluated their works on a single domain/dataset, so there is precedent in the community to evaluate multi-agent architectures in a single domain." - Multi-agent trajectory forecasting is a very different field today than it was just 2-3 years ago. There have (literally) been hundreds of papers published on the topic since 2018-2019 on a variety of problem domains. As a result, when submitting such a work to a general conference (like NeurIPS), it is important to understand if the approach is performant on only one dataset/domain or is able to maintain performance more generally (e.g., on 2 or 3 datasets/domains). This is not to say that a highly-specialized work cannot be considered a contribution, it is only to say that it could be more difficult for a wider audience to understand the key takeaways and leverage them in their own research, which is arguably one of the core goals of a broad conference such as NeurIPS.
> >
> > In summary, even if all other points are perfectly addressed in an update to the paper, the absence of comparisons on (large-scale) public datasets leads me to maintain my original rating.

---

### Official Review · Reviewer_ETWf · 2021-07-19

**Rating:** 6
**Confidence:** 4

**Summary:**

The manuscript presents a modification to the original transformer architecture that allows modeling (un-)ordered sequences of unordered sets. The TxT self-attention masks for sequences of T inputs in the original transformer are replaced by TxKxTxK mask tensors for a sequence of T input sets with K unordered entities each.
The architecture is used  in an autoregressive fashion for modeling the trajectories of basketball players and the ball. At each time-step the input consists of a set of players, represented by an identity embedding, raw x and y coordinates, and an optional context feature which indicates the direction of the player’s frontcourt. In task P, the set of target labels indicate the 2d motion corresponding to each input player. In Task B, the output labels indicate the 3d motion of the ball.


**Ethical Concerns:**

None.

**Limitations And Societal Impact:**

Limitations/Questions
1. No variance is reported for the experiments. Does this mean you evaluated a single run for each experiment? Without any estimate of the variance between random seeds, it is hard to get a stable estimate of relative performances of different models. Addressing this would take several more week-long experiments, which is maybe more a general issue with transformer architectures and their high computational demand.
1. “z” is used both for the z-axis in the ball position and the input sets.
1. Transformers are widely used, but a small general introduction or a figure of the overall architecture (maybe in the appendix) would make sense.
1. I think some details of the training setup are missing, e.g. what is the interval between validation steps. This is not a big problem though, since the code is made available.
1. DId you try to model sequences longer than 4s?


**Main Review:**

The proposed architecture addresses the interesting task of modeling trajectories of basketball players and ball during NBA games. Please find detailed comments in the sections below.

Strengths
1. The baller2vec model outperforms a strong graph-based recurrent neural network architecture, even with only a quarter of the training time.
1. I like the presentation of limitations, e.g. the discussion of the potential of overfitting to particular plays that can identify a team.
1. The network does not require the introduction of order among the players while also not relying on a GNN approach.
1. The authors present some good ablations and interesting observations, e.g. that the player embedding seems to cluster players with similar attributes (such as athleticism or high shooting ranges)


**Time Spent Reviewing:**

2

---

> ### Author Response · Authors · 2021-08-06
> **Thank You For Your Review!**
>
> Thank you for your review and for highlighting the strengths of our manuscript! We will address your other comments below.
>
> **No variance is reported for the experiments. Does this mean you evaluated a single run for each experiment? Without any estimate of the variance between random seeds, it is hard to get a stable estimate of relative performances of different models. Addressing this would take several more week-long experiments, which is maybe more a general issue with transformer architectures and their high computational demand.**
>
> Yes, training many runs using different random seeds was unfortunately computationally prohibitive for us. However, we will note that our model did not overfit during training (our best training loss was 0.495 compared to our test loss of 0.492), which might suggest variance would be low.
>
> **“z” is used both for the z-axis in the ball position and the input sets.**
>
> Thank you for bringing this to our attention. We will change the notation we use for the ball’s height.
>
> **Transformers are widely used, but a small general introduction or a figure of the overall architecture (maybe in the appendix) would make sense.**
>
> We will add a link to [“The Illustrated Transformer”](https://jalammar.github.io/illustrated-transformer/), which is an excellent introduction to Transformers.
>
> **I think some details of the training setup are missing, e.g. what is the interval between validation steps. This is not a big problem though, since the code is made available.**
>
> We evaluated our model on the validation set following each training epoch, which is somewhat implied in Line 184. Please let us know if you feel any other training details are missing.
>
> **DId you try to model sequences longer than 4s?**
>
> We did not, but exploring the effects of different temporal windows on model behavior would be an interesting research direction.

---

> > ### Comment · Reviewer_ETWf · 2021-08-25
> > **Thanks for your response.**
> >
> > After considering the other reviewer's comments which point out important limitations/issues, I decided to lower the rating to a 6.

---

### Official Review · Reviewer_d55H · 2021-07-19

**Rating:** 4
**Confidence:** 2

**Summary:**

This paper proposes a multi-entity generalization of the Tranformer architecture to address the problem of multi-agent spatiotemporal modeling (MASM). They demonstrate the performance of their model on two basketball-related tasks, and showcase the performance of their approach in comparison to standard baselines.



**Limitations And Societal Impact:**

See above.

**Main Review:**

The main contributions of the paper are as follows:
- they propose a generalization of the Transformer architecture to be suitable for multi-agent spatio-temporal modeling. To my knowledge, this is the first demonstratation of an approach which can simultaneously reaon about multiple agents in space-time.
- demonstration of efficacy of model on basketball dataset on 2 tasks (ball trajectory modeling, player trajectory prediction), comparing to standard baselines

Main concerns:
- Trajectory prediction and MASM are well-studied domains and it is somewhat surprising to see the use of naive baselines for comparison. It would have been prudent to compare the performance of their model against the most applicable SOTA methods
- Evaluation limited to a single dataset (basketball). I would be curious to see how this approach might perform on other datasets and related tasks (e.g. human trajectory prediction).

See for example:
“Pedestrian Trajectory Prediction Using Context-Augmented Transformer Networks”. This paper demonstrated the prediction results by using the Transformer network with two different information not only position data. At a high level, the method is conceptually similar to their mult-entity formalism.

Minor comment: Please justify why you'll have cast the prediction problem as a classification problem ?

Conclusion:
While the overall idea is interesting, the novelty of approach is not immediately clear and the evaluation is insufficient in its current forms. Comparisons to SOTA methods, other datasets, and generalization to other multi-agent problems would help strengthen the merits of the work.



**Time Spent Reviewing:**

2 hours

---

> ### Author Response · Authors · 2021-08-06
> **Regarding Our GRNN Baseline and Pedestrian Datasets**
>
> Thank you for your review and for highlighting the strengths of our manuscript! We will address your other comments below, and make the below points clearer in the manuscript.
>
> **Trajectory prediction and MASM are well-studied domains and it is somewhat surprising to see the use of naive baselines for comparison. It would have been prudent to compare the performance of their model against the most applicable SOTA methods**
>
> We compared `baller2vec` to a strong GRNN baseline (described starting at Line 191, with results in Table 2), which is directly inspired by Yeh et al. (2019), a SOTA model that was specifically evaluated on an NBA dataset, and thus very applicable. However, Yeh et al. (2019) did not include the hyperparameters of their architecture in their paper, did not publicly release their code, and did not reply to an email request for code/model hyperparameters, which made direct comparison to their approach impossible. Unlike Yeh et al. (2019) and many other papers that use sports datasets, the code for our entire pipeline, from data generation to model training, is public, which is an important contribution of our work to the community. We feel the lack of standardized benchmarks for sports datasets, combined with the general lack of open source code, significantly complicates comparing different models and has hindered the pace of research in this domain.
>
> **Evaluation limited to a single dataset (basketball). I would be curious to see how this approach might perform on other datasets and related tasks (e.g. human trajectory prediction).**
>
> Unfortunately, the commonly used pedestrian datasets (i.e., ETH, HOTEL, UNIV, ZARA1, and ZARA2) are *extremely* small, which makes them insufficient for evaluating an architecture that is known to thrive on large datasets (researchers designing Transformers for language modeling are not expected to evaluate their architectures on datasets only containing thousands of sentences). Specifically, whereas our NBA dataset has ~82 million different time steps in the training set, the total number of time steps for all of the previously mentioned pedestrian datasets combined is only 6,982.
>
> We also note that Social-BiGAT (Kosaraju et al., 2019), a NeurIPS 2019 paper [1], was exclusively evaluated on the previously mentioned pedestrian datasets, and Zhan et al. (2019), an ICLR 2019 paper [2], exclusively evaluated their model on an NBA dataset, so there is precedent in the community to evaluate multi-agent architectures in a single domain. However, we look forward to the opportunity to evaluate `baller2vec` on other large, multi-agent datasets when they become available.
>
> **Minor comment: Please justify why you'll have cast the prediction problem as a classification problem ?**
>
> We chose to cast the problem as a classification task to account for the multimodal nature of the agent trajectories at each time step (Line 107). To provide an example of this multimodality, a point guard with a defender directly in front of him might go to the left 50% of the time and to the right the other 50% of the time. While multimodality can be addressed with other modeling choices (e.g., using a mixture of Gaussians), the binned approach has interpretability benefits (e.g., easy to visualize) and also makes it easier to enforce physical constraints (e.g., the fact that a player cannot travel more than a certain number of feet in a time step).
>
> **While the overall idea is interesting, the novelty of approach is not immediately clear and the evaluation is insufficient in its current forms.**
>
> While we recognize that `baller2vec` is not a major technical innovation in terms of model components, we believe our reinterpretation of the attention mask matrix as a reshaped tensor is valuable because it allows the standard Transformer architecture to be applied to a broader class of problems (similar to ViT (Dosovitskiy et al., 2021)—an ICLR 2021 paper [3]). The fact that, in the 3.5 years between when the original Transformer manuscript and `baller2vec` were published, no other researchers had attempted to use a Transformer for multi-agent spatiotemporal modeling in this way suggests this approach was not a priori obvious. Further, [like Yoshua Bengio](https://www.facebook.com/100002848499232/posts/2805055846266004/), we believe that the application of existing technologies to challenging, real-world problems (MASM in our paper) contributes valuable knowledge to the community.
>
> **References**
>
> [1] Vineet Kosaraju, Amir Sadeghian, Roberto Martín-Martín, Ian Reid, Hamid Rezatofighi, and Silvio Savarese. Social-bigat: Multimodal trajectory forecasting using bicycle-gan and graph attention networks. In Advances in Neural Information Processing Systems, 2019.
>
> [2] Eric Zhan, Stephan Zheng, Yisong Yue, Long Sha, and Patrick Lucey. Generating multi-agent trajectories using programmatic weak supervision. In International Conference on Learning Representations, 2019.
>
> [3] Alexey Dosovitskiy, Lucas Beyer, Alexander Kolesnikov, Dirk Weissenborn, Xiaohua Zhai, Thomas Unterthiner, Mostafa Dehghani, Matthias Minderer, Georg Heigold, Sylvain Gelly, Jakob Uszkoreit, and Neil Houlsby. An image is worth 16x16 words: Transformers for image recognition at scale. In International Conference on Learning Representations, 2021.

---

### Public Comment · ~Michael_A._Alcorn1 · 2021-11-20
**Datasets in Other Multi-Agent Trajectory Papers Accepted to NeurIPS 2021**

For posterity, I want to note that at least two other multi-agent trajectory modeling papers that were accepted to NeurIPS this year only evaluated their methods on a single, large real-world dataset. "[GRIN: Generative Relation and Intention Network for Multi-agent Trajectory Prediction](https://openreview.net/forum?id=ephWA7KaWmD)" evaluated their method on a small simulated dataset (50K training sequences) and a preprocessed NBA dataset (100K training sequences). "[Collaborative Uncertainty in Multi-Agent Trajectory Forecasting](https://openreview.net/forum?id=sO4tOk2lg9I)" evaluated their method on the nuScenes dataset (1,000 scenes) and Argoverse (206K training sequences). Additionally, the 2020 NeurIPS paper "[Learning Agent Representations for Ice Hockey](https://proceedings.neurips.cc/paper/2020/hash/d90e5b6628b4291225cba0bdc643c295-Abstract.html)" was only evaluated on a hockey dataset.

---

### Public Comment · ~Michael_A._Alcorn1 · 2024-01-31
**Good enough for Waymo/ICCV**

Researchers from Waymo used almost exactly the `baller2vec` architecture (compare Figures 1 and 8 from the Waymo paper with Figures 2 and 4 from `baller2vec`) to achieve state-of-the-art results in the self-driving car setting in the ICCV 2023 paper "[MotionLM: Multi-Agent Motion Forecasting as Language Modeling](https://openaccess.thecvf.com/content/ICCV2023/html/Seff_MotionLM_Multi-Agent_Motion_Forecasting_as_Language_Modeling_ICCV_2023_paper.html)".

---

### Decision · Program_Chairs · 2021-09-27

**Decision:**

Reject

**Comment:**

The reviewers generally held the opinion that the experimental evaluation of this method was insufficient for acceptance.
Multiple reviewers cited the lack of comparison to other methods. Multiple reviewers brought up the issue of not reporting variances for model predictions and this issue was not adequately addressed. Reviewers also pointed to a lack of comparisons with other high performing methods and comparison of this method on other datasets as being outstanding issues.

As a result of these issues only one of four reviewers gave a final recommendation of acceptance and the ACs final recommendation is to not accept this paper.

While one of the reviewers had previously reviewed an older version of this manuscript at ICML, they were careful in their review of this submission and their opinion was not a decisive factor in making the final decision for this paper.